# A Systems Biology Approach for Prioritizing ASD Genes in Large or Noisy Datasets

**DOI:** 10.3390/ijms26052078

**Published:** 2025-02-27

**Authors:** Veronica Remori, Heather Bondi, Manuel Airoldi, Lisa Pavinato, Giulia Borini, Diana Carli, Alfredo Brusco, Mauro Fasano

**Affiliations:** 1Department of Science and High Technology, University of Insubria, 22100 Como, Italy; vremori@uninsubria.it (V.R.); heather.bondi@uninsubria.it (H.B.); mairoldi@uninsubria.it (M.A.); 2Department of Medical Sciences, University of Torino, 10126 Torino, Italy; lisa.pavinato@unito.it (L.P.); diana.carli@unito.it (D.C.); 3Institute of Oncology Research (IOR), Bellinzona Institutes of Science (BIOS^+^), 6500 Bellinzona, Switzerland; 4Faculty of Biomedical Sciences, Università della Svizzera Italiana, 6900 Lugano, Switzerland; 5Department of Public Health and Pediatric Sciences, University of Torino, 10126 Torino, Italy; giulia.borini@edu.unito.it; 6Department of Neuroscience “Rita Levi Montalcini”, University of Torino, 10126 Torino, Italy; alfredo.brusco@unito.it; 7Neuroscience Research Center, University of Insubria, 21052 Busto Arsizio, Italy

**Keywords:** autism, protein–protein interaction network, betweenness centrality, systems biology, gene prioritization

## Abstract

Autism spectrum disorder (ASD) is a complex multifactorial neurodevelopmental disorder. Despite extensive research involving genome-wide association studies, copy number variant (CNV) testing, and genome sequencing, the comprehensive genetic landscape remains incomplete. In this context, we developed a systems biology approach to prioritize genes associated with ASD and uncover potential new candidates. A Protein–Protein Interaction (PPI) network was generated from genes associated to ASD in a public database. Leveraging gene topological properties, particularly betweenness centrality, we prioritized genes and unveiled potential novel candidates (e.g., *CDC5L*, *RYBP*, and *MEOX2*). To test this approach, a list of genes within CNVs of unknown significance, identified through array comparative genomic hybridization analysis in 135 ASD patients, was mapped onto the PPI network. A prioritized gene list was obtained through ranking by betweenness centrality score. Intriguingly, by over-representation analysis, significant enrichments emerged in pathways not strictly linked to ASD, including ubiquitin-mediated proteolysis and cannabinoid receptor signaling, suggesting their potential perturbation in ASD. Our systems biology approach provides a promising strategy for identifying ASD risk genes, especially in large and noisy datasets, and contributes to a deeper understanding of the disorder’s complex genetic basis.

## 1. Introduction

Autism spectrum disorder (ASD) is a complex multifactorial neurodevelopmental disorder involving many genes [1]. It comprises several childhood diseases with deficits in social communication and interaction. The prevalence of ASD is about 1% in the general population [2]. The role of de novo mutations in ASD has been well established, and several known ASD-causing genes are listed in the Simons Foundation Autism Research Initiative (SFARI) Gene database [3], which classifies genes as syndromic [4] or non-syndromic and assigns three categories of score according to their confidence level (score 1—high confidence, score 2—strong candidate, score 3—suggestive evidence, according to the presence of de novo likely-gene-disrupting mutations reported in the literature). However, the set of genes implicated to date is still far from complete.

To unravel the genetic complexity of ASD, several large-scale genome-wide association studies (GWASs), copy number variant (CNV) testing, and genome sequencing have been performed, resulting in the identification of many candidate variants [5,6]. Among these techniques, array-comparative genomic hybridization (array-CGH) is the molecular karyotyping technique of choice that allows the investigation of unbalances in the copy number of genes (deletions, duplications, or triplications) [7]. While these approaches yield large datasets, such as those from whole-exome sequencing (WES), they also present significant challenges, including the need to manage vast amounts of information and the difficulty of accurately identifying pathogenic variants. Additionally, CNV data represent a noisy dataset due to variability in resolution, detection thresholds, and the inclusion of variants of uncertain significance (VUSs). This inherent noise complicates the process of prioritizing truly relevant genes, highlighting the need for robust methods capable of filtering and ranking candidates within these complex datasets.

In this context, ASD may be considered predominantly as a multifactorial disorder, resembling a complex system where the components are proteins connected by physical or functional interactions [8]. A Protein–Protein Interaction (PPI) network is a graph in which proteins serve as nodes and physical interactions are represented by edges. PPIs are biologically meaningful, since many functionally cooperating proteins form complexes and often carry out their function through interactions with other proteins [9]. Physical interactions can be queried in multiple databases, including the International Molecular Exchange Consortium (IMEx) database [10]. This public database emerged from an international collaboration above major public interaction data providers, who have agreed to collaborate on curation effort. It comprises experimental validation of interactions documented within a publication, detailing aspects such as host organism, assay methods, and constructs involved. Interactions are defined broadly, including protein–protein, protein–small molecule, and nucleic acid interactions, with guidelines for transcription factor–gene interactions and downstream effects like gene expression regulation [11].

The topological analysis of a PPI network is a valuable tool for the discovery of key players in a disease network model. Centrality measures such as closeness centrality or betweenness centrality may lead to the identification of potential pharmacological targets or candidate genes to be included in the genome sequencing pipeline, e.g., variant calling [12]. PPI networks may be further analyzed in terms of pathways to show how protein clusters are functionally linked to specific biological processes [13]. Pathway analysis is employed to identify activated metabolic or signaling mechanisms from omics data, considering the actual functional actors involved in regulatory pathways [14], and it is usually performed by over-representation analysis (ORA). ORA is a statistical method used to highlight relevant pathways and provide functional explanation for changes in protein abundance revealed by differential quantitative proteomics [15]. ORA determines whether genes from pre-defined datasets are present more than would be expected in a subset of data. It assumes that relevant pathways can be detected if the ratio of differentially expressed genes exceeds the ratio of genes randomly expected within a given pathway or biological process. The probability value for the null hypothesis is typically computed by the Fisher exact test with Benjamini–Hochberg multiple-testing correction [16].

Here, we developed a systems biology approach to prioritize genes related to ASD and discover possible new candidate genes starting from a list of genes related to ASD in the SFARI database. As a proof of concept, we mapped genes within CNVs identified in ASD patients to our network model to obtain a prioritized gene list.

## 2. Results

### 2.1. SFARI-Based Network Model

The SFARI database was queried to gather non-syndromic genes with SFARI score 1 and 2 (117 score 1, 651 score 2, 768 genes in total). Utilizing the IMEx database, the first interactors of SFARI genes were retrieved. The resulting list was used to generate a PPI network (network A) formed by 286,266 edges and 12,598 nodes (Figure 1a, Appendix A). It included 224 score 1 genes, 696 score 2 genes, and 101 score 3 genes, with 252 syndromic genes and 845 non-syndromic genes. Notably, network A was strongly enriched in SFARI genes, since they represented 96.5% of score 1 genes, 98.9% of score 2 genes, and 82.8% of score 3 genes annotated in the SFARI database. Considering the high number of nodes in the network, representing approximately 63% of all protein-coding genes in the human genome, we evaluated whether the enrichment in SFARI genes was significantly associated with network A or if a comparable enrichment could be observed by randomly selecting proteins in a number equal to the network size to assess the specificity of the network. To check this, 12,598 protein-coding genes were randomly selected from the HGNC database using 1000 random seeds. These randomly generated lists comprised 46.6 ± 2.1% score 1 genes, 56.2 ± 1.6% score 2 genes, and 36.7 ± 2.4% score 3 genes from the SFARI database (Appendix A). We observed statistically significant differences between the percentages of SFARI genes in network A and those in the randomly generated lists (*p*-value < 2.2 × 10^−16^; one-sample *t*-test). To evaluate the sensitivity of network A in retrieving brain expressed genes, we compared the expression data in the brain from the Human Protein Atlas database with our list of nodes, assessing that 11,879 genes were expressed in at least one brain area, corresponding to 94.3% of the whole network. Subsequently, we extracted the 1097 SFARI genes present in network A. Among them, 80.5% genes/nodes showed physical interactions. Indeed, only 214 nodes resulted as unconnected, whereas 875 nodes were organized in a highly connected network, and four node pairs were present (Figure 1b). Topological analysis showed that only a few nodes were highly connected, as expected in an undirected biological network (Appendix A). Also, it allowed us to rank gene symbols by decreasing betweenness centrality, and the 30 most central genes were reported in Table 1. Note that betweenness centrality was correlated with all the other topological metrics (Appendix A).

Gene expression data from 13 brain regions, as reported in the Human Protein Atlas, were analyzed to assess the relevance of the identified genes in the context of ASD. The correlation between the expression levels of the top 30 genes (Table 1) and the 224 score 1 genes present in network A was then calculated. A strong correlation was observed between the SFARI genes and the top 30 genes, indicating that these genes are co-expressed in key brain regions (Figure 2).

### 2.2. Validation on ASD Patients

A cohort of 466 both syndromic and non-syndromic ASD patients underwent array-CGH analysis (78% males; 22% females; age 3–59 years, mean age 19.2 ± 12.6). Among them, 177 rare CNVs were identified, 157 of which were classified as variants of uncertain significance (VUSs) according to the ACMG criteria (PMID: 31690835), with 61 deletions and 96 duplications. The mean size of deletions was 449 kb ± 734 kb (range: 20–3390 kb), while the mean size of duplications was 450 kb ± 734 kb (range: 31–7750 kb). Eighty-one CNVs were not segregated, and seventy-two were inherited; however, a comprehensive psychiatric evaluation of the parents was not always available, and four were de novo. The list of DECIPHER IDs for the 135 patients carrying the 157 CNVs classified as VUSs and included in the validation cohort is available in Appendix A. Overall, 768 genes were localized in CNVs classified as VUSs. To identify and prioritize genes potentially related to ASD, we mapped these genes in network A and ranked them according to their betweenness centrality score (Table 2 and Appendix A). Only 456 genes were mapped, since the CNV list included not only protein-coding genes but also pseudogenes and non-coding sequences. Among the mapped genes, twelve genes were classified as SFARI score 1, fifty-six as score 2, and seven as score 3. Additionally, 10 were syndromic and 68 were non-syndromic genes. From a network perspective, mapped genes were organized in several PPI networks: a main component of one hundred eighty-seven nodes (network B), and a secondary component with six nodes, nine triples, thirteen pairs, and two hundred ten single nodes (Figure 3).

To elucidate the potential roles of the mapped genes on known biological and pathogenic pathways, we performed ORA (Table 3). We observed significant enrichments in developmental syndromes, specifically 22q11.2 deletion syndrome, and Prader–Willi and Angelman syndromes. Additionally, there was a significant enrichment in the cannabinoid receptor signaling.

To further explore pathways linked to genes holding central roles in network A, we generated three sublists of genes encompassing the top 80%, 50%, and 20% of genes, based on their betweenness centrality measures. In the top 80% gene list (364 genes), significant enrichments were observed in 22q11.2 deletion syndrome, Prader–Willi and Angelman syndromes, and cannabinoid receptor signaling terms. Focusing on the top 50% of genes (228 genes), we found significant enrichments in 22q11.2 deletion syndrome, Prader–Willi syndrome, and Angelman syndrome. Notably, the ubiquitin-mediated proteolysis pathway also showed significant enrichment.

When selecting the most central nodes (top 20%, 91 genes), 22q11.2 deletion syndrome, Prader–Willi syndrome, and Angelman syndrome remained significantly enriched. Notably, FCERI-mediated MAPK activation and Neutrophil degranulation resulted significantly enriched. To ensure the robustness of ORA results in relation to node centrality, we generated three random lists containing the same number of genes as the previous lists (364, 228, and 91 genes, respectively). These genes were randomly selected from the list of mapped genes. Subsequently, ORA was conducted on these three random lists. We observed a significant enrichment in 22q11.2 deletion syndrome across all randomly sampled lists, and Prader–Willi and Angelman syndromes showed significant enrichments in the 364-gene list. The persistent enrichment of these pathways in the random-ORA suggests an inherent association with this set of genes.

Next, we performed the ORA in network B (Table 4). Here, we observed a significant enrichment of pathways already identified in the previous analysis (i.e., 22q11.2 deletion syndrome, Prader–Willi and Angelman syndromes, ubiquitin-mediated proteolysis, FCERI-mediated MAPK activation). Furthermore, Salmonella infection, progesterone-mediated oocyte maturation, RNA polymerase, and tight junction resulted significantly enriched.

## 3. Discussion

We employed a systems biology approach to build a model of ASD-related genes and their physical interactions. Starting from an input list of SFARI genes annotated with score 1 or score 2, non-syndromic, we generated a network highly enriched in ASD-related genes, as confirmed by the Monte–Carlo validation. Among them, 94.3% of genes are expressed in the brain, according to the Human Protein Atlas database. In order to maintain the possibility of identifying novel genes which still are not annotated with expression data, we did not apply any brain expression filter to the network. Indeed, there may be genes expressed in the brain that have not yet been identified and could still play a crucial role in ASD.

Furthermore, the topological analysis of network A allowed us to rank all genes according to their betweenness centrality. Betweenness centrality is a global centrality metric that measures the influence of a node over the spread of information in a network. In a biological network context, it facilitates the identification of proteins involved in the spread of perturbations across the network. Indeed, proteins with high betweenness centralities are referred to as key connector proteins with crucial functional and dynamic features [17]. However, we acknowledge that proteins with high-centrality nodes are not necessarily disease genes. Nevertheless, these nodes and their interactors may play a significant role in propagating perturbations originating from disease genes. In this view, these genes do not necessarily carry a variant themselves, but they could still play a critical role in disease manifestation by influencing the network dynamics and facilitating the propagation of pathogenic signals. Table 1 showed the 30 most central genes: among them, *DISC1*, *CUL3*, and *HRAS* were input genes present in the SFARI database as strong candidate (*DISC1*) or high confidence (*CUL3*, *HRAS*) genes related to ASD. Moreover, *YWHAS*, *YWHAZ*, and *MAPT* genes are listed in the SFARI database as category 3 genes, which have a single reported de novo likely-gene-disrupting mutation. The most central protein-coding gene is *ESR1*, which was annotated in the database of gene-disease association DisGeNET [18] as a gene related to ASD. It encodes an estrogen receptor and ligand-activated transcription factor, and it could be related to ASD and to severities in the impairment of social interaction and emotional regulation [19,20]. However, OMIM does not correlate *ESR1* with a neurologic phenotype yet. *LRRK2*, *APP*, and *HTT* genes are primarily known for their involvement in relevant neurodegenerative diseases, as highlighted by OMIM annotations. Interestingly, several clinical studies suggest that haploinsufficiency of *LRRK2* may also play a role in neurodevelopmental disorders [21]. Moreover, *APP* and *HTT* are also annotated in the DisGeNET database as associated with ASD. In our ranked list, three genes (*ATXN1*, *FGFR3*, and *HSBP1)* are strongly associated with rare neurological diseases according to OMIM annotations and have been suggested to be related to ASD [22,23,24]. Furthermore, other genes identified in our analysis, including *EGFR*, *JUN*, *CFTR*, *BMI1*, *GRB2*, *HSP90AB1*, *SNW1*, *CANX*, *NEK4*, and *ERBB2*, were suggested as related to ASD or causative of propagating a perturbation starting from an ASD disease gene according to the literature [25,26,27,28,29,30,31,32,33,34,35]. It should be noted that our analysis highlights not only genes that may be involved directly in ASD onset but also genes that may be perturbed as a consequence of variants on other ASD-linked genes. For instance, in our analysis, we found *CFTR*, which is linked to Cystic Fibrosis (OMIM #219700); we may hypothesize that this gene is not a main actor in ASD onset, but it may be perturbed by other genes that are involved in neurodevelopmental disorder pathways. In a similar way, this model allowed us to suggest some candidate genes that were not already associated to ASD, neither in DisGeNET nor in the literature, such as *CDC5L*, *RYBP*, and *MEOX2*. The mesenchyme homeobox gene, *MEOX2*, encodes a family of homeodomain transcription factors expressed within the vascular system. Traditionally, deficiencies in cerebrovascular networks are recognized as an underlying mechanism for altered cerebral blood flow and blood–brain barrier breakdown in neurodegenerative conditions [36]. Notably, recent preclinical evidence reveals a novel association between ASD and neurovascular abnormalities [37]. CDC5L, a DNA-binding protein crucial for cell cycle control, together with PRP19α constitutes the major components of the active spliceosome. Intriguingly, PRP19α plays a role in neuronal differentiation in cellular models [38]. Thus, it is plausible to posit that PRP19α may also play a role in neuronal differentiation and ASD. The zinc finger protein RYBP represents an essential component of the Polycomb repressive complex 1 (PRC1), contributing to repressive epigenetic factors. RYBP is found to regulate embryonic neurogenesis and neuronal development by modulating Notch signaling, both in vivo and in vitro [39].

To further investigate the relevance of these identified genes in the context of ASD, we analyzed gene expression data from 13 brain regions available in the Human Protein Atlas. Our findings showed that the expression of the top 30 genes identified in the betweenness centrality analysis correlates highly with the 224 SFARI score 1 genes, suggesting that these genes are co-expressed in key brain regions relevant to ASD. However, we also noted that the gene *CUL3*, despite being a score 1 gene, showed low correlation with other SFARI genes, highlighting an important aspect of gene interaction within the context of ASD. This underscores that while co-expression is a valuable tool for understanding gene function, it should not be the sole criterion for prioritizing genes in ASD research. Even in the absence of co-expression, genes with high betweenness centrality may play critical roles in the regulation of ASD-related pathways. Indeed, the PPI network analysis provided an additional level of insight into the relevance of these genes. By focusing on genes with high betweenness centrality, we were able to identify genes that occupy central positions in the network, linking various functional modules together. These genes may not necessarily exhibit co-expression with other ASD-related genes, yet their central roles in regulating molecular interactions suggest they are crucial for the modulation of ASD-related pathways.

The PPI network approach proved particularly valuable in cases where gene expression data were unavailable or incomplete. By integrating PPI data with genetic information, we were able to offer a more comprehensive framework for identifying potentially important genes in ASD pathogenesis, even from datasets that may otherwise be challenging to interpret. As a proof of concept, we extended our SFARI-based network model to prioritize genes with VUSs contained in CNVs in ASD patients. The use of CNV data in this study is particularly significant, as CNVs represent a prototypical noisy dataset, where variability in resolution and detection thresholds often result in high false-positive rates. By mapping this list of genes on network A, we obtained a list of genes of interest ordered according to their betweenness centrality. Among the 30 most central genes, ten protein-coding genes are annotated in the SFARI database (*RAC1*, *PRKN*, *MAPK3*, *KCNMA1*, *UBE3A*, *AKAP9*, *HERC2*, *CHR7*, *ZMYND11*, and *GABRB3*). Thus, 33% of the top 30 genes are SFARI genes. Note that the proliferation marker protein Ki-67 expressed by *MKI67* was classified as associated to ASD in the DisGeNET database, whereas no neurological phenotype was associated in OMIM. Also, the literature associated Ki-67 with immune dysregulation, since increased Ki-67 production/expression was typically seen in children with ASD [40]. Moreover, *SMARCB1* encoded for the core component of the BAF (hSWI/SNF) complex which played important roles in cell proliferation and differentiation, and it was present in the DisGeNET database as related to ASD. It was believed to be causative for Kleefstra syndrome spectrum (KSS), a neurodevelopmental disorder with clinical features of both intellectual disability (ID) and autism spectrum disorder (ASD) [41]. In addition, *FGFR2* and *COMT* were also annotated in the DisGeNET database as associated to ASD. The ubiquitous fibroblast growth factor receptor (FGFR) type 2, a tyrosine kinase receptor involved in several biological activities, is encoded by the gene *FGFR2*. Numerous overlapping disorders are known to be caused by the mutant form of this protein, and it is suggested to also be involved in the genesis of ASD [42]. Instead, the enzyme catechol-O-methyltransferase (COMT) is essential for the regulation of catechol-dependent processes, including pain perception, heart function, and thought processes. It is one of several enzymes that degrade and inactivate catecholamine neurotransmitters, including dopamine, epinephrine, and norepinephrine. Since it is hypothesized that ASD may have a dopaminergic system malfunction in the midbrain as part of its pathophysiology, the role of *COMT* in ASD has been extensively studied [43]. On the other hand, *PARKAB2*, *F2RL1*, *HAX1*, and *PTP1B* are related to ASD according to the literature [44,45,46,47]. All other genes found from the mapping on network A are potentially linked to ASD and should be prioritized in their analysis.

To assess the correlation between these mapped genes and known pathogenic pathways, we performed over-representation analysis (ORA) on the genes that were identified by mapping the CNV list on network A. Unsurprisingly, we found significant enrichments in developmental syndromes (i.e., 22q11.2 deletion syndrome and Prader–Willi and Angelman syndromes). We acknowledge that these pathways may appear to reflect general neurodevelopmental syndromes, but they are consistent with the known genetic overlap between ASD and these developmental disorders. The observed enrichment in pathways linked to ASD supported our hypothesis that mapping genes from such a noisy dataset onto network A could be a valuable resource to prioritize potentially relevant variants even in a noisy dataset whereby expression data are unavailable. Additionally, we ranked the genes according to their betweenness centrality values and conducted ORA on the top 80%, 50%, and 20% of genes (364, 228, and 91 genes, respectively). Also, to ensure the reliability of our results, we generated three random lists containing the same number of genes as the previous lists, as explained in the Results Section. This random sampling approach was used to control for potential dataset biases, ensuring that the observed pathway enrichment is not a result of the methodology itself but rather reflects inherent associations in the dataset. The signature associated with the del22q11.2 syndrome is present in the whole list and in all sublists. However, the same pathway is enriched also when genes are randomly resampled, suggesting an inherent association with this set of genes. The syndrome associated with the 22q11.2 deletion is a rare CNV causing a congenital malformation disorder. The clinical phenotype includes cardiac defects, palatal anomalies, facial dysmorphism, developmental delay, and immunodeficiency [48].

Furthermore, the signature of Prader–Willi and Angelman syndromes is significantly enriched with protein-coding genes present in all sublists and in the list of 364 resampled genes. This syndrome is related to deletion of parental copies within the chromosome region 15q11-q13; more in detail, Prader–Willi syndrome (PWS) results from the deletion of the paternal copies of the imprinted *SNRPN* gene, the *NDN* gene, and possibly other genes, while Angelman syndrome (AS) is caused by de novo maternal deletions involving chromosome 15q11.2-q13 (70%), by paternal uniparental disomy of 15q11.2-q13 (2%), by imprinting defects (2–3%), and by mutations in the gene encoding the ubiquitin–protein ligase *UBE3A* gene (25%) [49]. Note that Prader–Willi and Angelman syndromes are suggested to be related to ASD. For instance, AS is caused by mutations in the *UBE3A* gene, which is implicated in neurodevelopmental disorders [50]. In addition, ASD has been linked with maternally derived duplications/triplications of chromosome 15q11-13 [51].

Also, the signature of ubiquitin-mediated proteolysis is present in the top 50% genes list. The ubiquitin proteolytic system plays a key role in the regulation of the cell cycle, the modulation of the immune and inflammatory responses, the control of signal transduction pathways, development, and differentiation. These processes are controlled via degradation of a specific protein or a subset of proteins [52]. For example, mutations in *HERC2*, which is a large E3 ubiquitin ligase with multiple structural domains, are linked to developmental delays and impairment caused by nervous system dysfunction.

Moreover, the signature of the cannabinoid receptor signaling pathway is present in the whole list and in the sublist formed by the top 80% of genes mapped on network A. The endocannabinoid system (ECS) plays a key modulatory role in synaptic plasticity and homeostatic processes in the brain [53]. The ECB system is linked to four phenotypic features known to be atypical in ASD (social reward responsivity, neural development, circadian rhythm, and anxiety-related symptoms) [54]. Thus, there are studies suggesting that CBD-rich cannabis may yield benefits for some individuals with ASD [55,56].

Note that the pathway of the high-affinity IgE receptor (Fc epsilon RI, or FCERI)-mediated MAPK activation is significantly enriched for the 20% sublist. More in detail, the formation of the LAT signaling complex leads to activation of MAPK and production of cytokines. The mitogen-activated protein kinase (MAPK) pathways are intracellular signaling pathways necessary for the regulation of several physiological processes, including neurodevelopment [57].

The ORA performed on genes in network B did not add any relevant insights with respect to the ORA performed on the whole list of CNV genes mapped on network A. Indeed, 22q11.2 deletion syndrome and Prader–Willi and Angelman syndromes are represented in the dataset with a high number of genes. Consequently, they are found with a low FDR in both sets of genes.

## 4. Materials and Methods

### 4.1. PPI Network Generation Workflow

The input list of genes associated with ASD was downloaded from https://gene.sfari.org/ (accessed on 13 November 2023; 10-2023 release) and used to generate a network encompassing all first interactors of the gene products. Cytoscape 3.9.1 [58] was used to generate a PPI network, starting from score 1 and 2 non-syndromic SFARI genes. The IMEx consortium database was queried through Cytoscape using the PSICQUIC standard (Proteomics Standard Initiative Common QUery InterfaCe) created by the Human Proteome Organization Proteomics Standards Initiative (HUPO-PSI) [59]. The network was filtered for taxonomy ID 9606 (*Homo sapiens*) to remove homology inferences, and the edge table and the node table were processed using a script in R to replace the UNIPROT IDs with the corresponding gene symbols to remove redundancy due to possible aliases in UNIPROT accession numbers in the IMEx database. Also, duplicated edges and self-loops were removed. A list of all nodes in the present network (input list and first interactors) was used to query the IMEx database again to identify second interactors and interactions among first interactors. The edge and the node tables were processed, as previously described. Subsequently, second interactors were removed to consider only genes interacting with the input list made of score 1 and 2 non-syndromic SFARI genes. This process led to the identification of new edges connecting the first interactors.

Subsequently, the enrichment in SFARI genes was assessed using a Monte–Carlo approach with one-sample *t*-test. Briefly, we randomly sampled the same number of genes from the Human Genome Nomenclature Committee (HGNC) database using 1000 different random seeds. We computed the average number of SFARI genes in this random population together with its standard deviation, and we calculated the *p*-value for the real number of SFARI genes with respect to the null population, i.e., the distribution of the number of SFARI genes in the random lists.

Furthermore, we evaluated the expression in the brain of the genes present in the network by incorporating gene expression data from the Human Protein Atlas (https://www.proteinatlas.org/humanproteome/brain/data, Version 24.0, accessed on 20 February 2025). Specifically, we filtered interactions based on gene expression levels in 966 samples from the Human Brain Tissue Bank (HBTB), which were analyzed using RNA-seq on the MGI platform.

### 4.2. Topological Analysis of the Network

To classify all genes, a topological analysis was performed. The betweenness centrality was calculated for each node by using the NetworkAnalyzer utility in the Cytoscape Environment. The betweenness centrality considers how many geodesic paths cross a given node *n*_i_ among all geodesic paths in a network. This metric is moderated by the total number of shortest paths existing between any couple of nodes of the network. Nodes that lie on a larger number of shortest paths have higher betweenness centrality scores. The betweenness centrality of node *n_i_* is given by the following:$$C_{B}\left(n_{i}\right)=\sum_{j<k}\frac{\sigma_{jk}\left(n_{i}\right)}{\sigma_{jk}}\frac{1}{\left(\begin{array}{c} N\\ 2 \end{array}\right)}$$
where *N* is the total number of nodes, *σ_jk_* is the number of geodesics between nodes *n_j_* and *n_k_*, and *σ_jk_*(*n_i_*) is the number of geodesic pathways containing node *n_i_* [12].

### 4.3. Gene Expression Analysis in ASD-Related Brain Regions

To assess the relevance of the identified genes in the context of ASD, we analyzed gene expression data from 13 brain regions available in the Human Protein Atlas. These regions include the amygdala, basal ganglia, cerebellum, cerebral cortex, choroid plexus, hippocampal formation, hypothalamus, medulla oblongata, midbrain, pons, spinal cord, thalamus, and white matter. We then computed the correlation between the expression levels of the top 30 genes of the PPI network ranked by betweenness centrality and the genes associated with high-confidence autism-related traits (SFARI score 1) present in the PPI network.

### 4.4. Array-Comparative Genomic Hybridization

Patients with a neurodevelopmental disorder (NDD) were enrolled at different hospitals in diagnostic or research projects. Informed consent was obtained from the participating families, and the study protocol was approved by the relevant Ethics Committee, according to the Declaration of Helsinki. A cohort of 466 both syndromic and non-syndromic ASD patients underwent array-CGH analysis (SurePrint G3 Human CGH Microarray Kit, 8 × 60 K, CytoGenomics software 5.4 microarray data analysis; Agilent Technologies, Santa Clara, CA, USA). The DECIPHER database (version 11.29) was used as the reference for the classification of CNVs according to the 2020 ACMG criteria. One hundred thirty-five patients had one or more CNVs for a total of one hundred fifty-seven classified as VUSs. For our analysis, the genes localized in each deleted or duplicated region classified as VUSs were listed, generating a catalogue of 768 genes, hereafter indicated as the CNV gene list.

### 4.5. Over-Representation Analysis

Over-representation analysis (ORA) was performed on the CNV gene list mapped on the PPI network using the WEB-based GEne SeT AnaLysis Toolkit (WebGestalt) 2019 [60]. As the functional databases, Reactome (09-2018 release, version 66), Kyoto Encyclopedia of Genes and Genomes (KEGG) (01-2018 release, version 88.2), and Wikipathway (10-2020 release) were selected, and “human genome, protein coding” was used as the reference set. Enriched categories with a Benjamini–Hochberg false discovery rate (FDR) ≤ 0.05 were considered statistically significant.

Additionally, mapped genes were ranked by betweenness centrality values, and ORA was performed on the top 80%, 50%, and 20% of genes. This strategy allowed for the identification of pathways that are associated with genes holding central roles in the network. Moreover, to discern pathways associated with the centrality value of nodes in the network, rather than the inherent nature of the dataset (dataset bias), three different gene lists were randomly generated, each with the same number of genes selected from the initial set of mapped genes on the network. This random sampling was repeated three times to perform the random ORA in triplicate. Pathways that showed significant enrichment in at least two out of three random ORAs were considered to be due to biases within the input gene list.

## 5. Conclusions

In summary, we generated a PPI network using genes annotated in the SFARI database as associated with ASD. By leveraging the topological properties of nodes in the network, such as betweenness centrality, we prioritized genes related to ASD and suggested possible novel candidates. Noteworthy, among the highly central nodes, *CDC5L*, *RYBP*, and *MEOX2* genes were identified, and they may be proposed as potential candidates for ASD. Furthermore, we expanded the application of our method as a novel prioritization strategy for variant calling, ranking CNV genes according to their betweenness centrality value. Additionally, through ORA, pathways not strictly linked to ASD emerged, including the ubiquitin-mediated proteolysis and the cannabinoid receptor signaling, uncovering new mechanistic insights within the molecular background of ASD. These findings highlight how our prioritization strategy is capable of revealing novel associations despite the inherent noise in CNV datasets.

One of the main limitations of this study is that the generation of the PPI network heavily relies on interaction data retrieved from available databases, such as IMEx, in the context of this work. These databases are inherently biased, as they tend to contain more interaction data for proteins that have been extensively studied, while proteins that have attracted less research attention often have a less characterized interactome. This can potentially limit the comprehensiveness of the proposed PPI network, as certain interactions might be underrepresented or entirely missing. Similarly, the ORA is also influenced by biases present in pathway databases. Certain pathways are more frequently represented simply because they have been the focus of more research, while other pathways may be underreported. This can lead to an overemphasis on well-studied pathways and may limit our ability to identify novel or less-characterized pathways that could be relevant to ASD. Moreover, pathways are intrinsically redundant, because the same genes may belong to distinct signatures.

Despite these limitations, we believe that our systems biology approach is particularly effective in noisy datasets and provides a promising strategy for identifying ASD risk genes and contributes to a deeper understanding of the genetic basis of ASD. In the future, we will apply our model on several datasets from exome sequencing experiments.

## Figures and Tables

**Figure 1 ijms-26-02078-f001:**
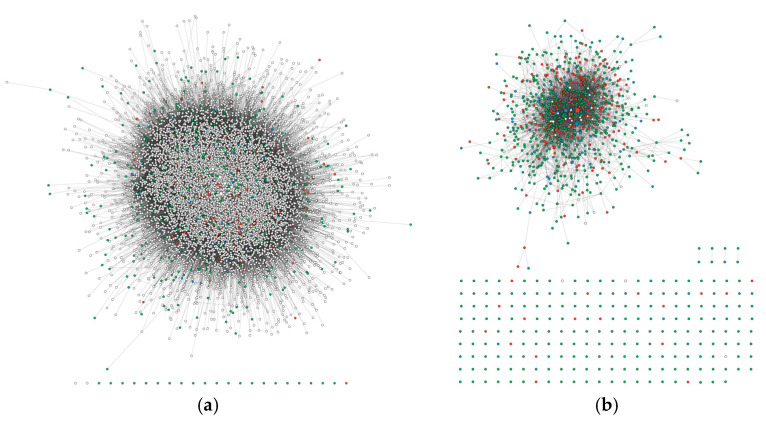
(**a**) PPI network of score 1 and 2 SFARI genes and their first interactors. Red: score 1, green: score 2, blue: score 3. The node size is proportional to betweenness centrality. (**b**) A subset of SFARI genes. Red: score 1, green: score 2, blue: score 3, white: syndromic without score. The node size is proportional to betweenness centrality.

**Figure 2 ijms-26-02078-f002:**
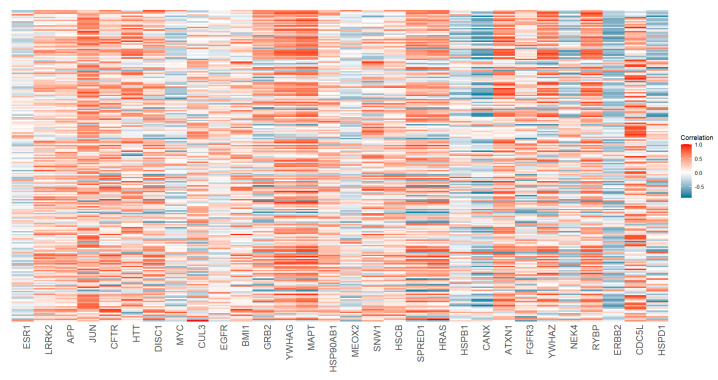
Heatmap showing the correlation between the top 30 identified genes (x-axis) and the 224 high-confidence autism-related genes (SFARI score 1, y-axis). Correlation values are represented by a color gradient, with blue indicating negative correlations, red indicating positive correlations, and white representing no correlation.

**Figure 3 ijms-26-02078-f003:**
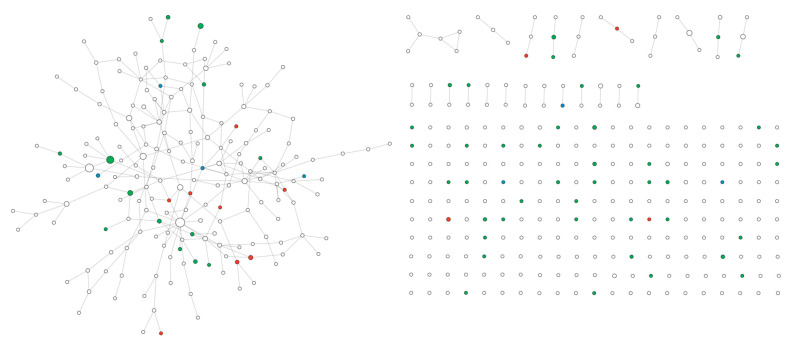
PPI network of genes within CNVs mapped on network A. Red: score 1, green: score 2, blue: score 3. The node size is proportional to betweenness centrality.

**Table 1 ijms-26-02078-t001:** The top 30 genes ranked by betweenness centrality in the PPI network originated from score 1 and 2 SFARI genes (network A).

Gene	SFARI Score	Syndromic	Betweenness Centrality	Relative Betweenness Centrality ^a^ (%)	Expression in Brain ^b^ (TPM)	Brain Expression ^c^	OMIM Phenotype ^d^
*ESR1*			0.0441	100	1.334	low	
*LRRK2*			0.0349	79.14	4.878	low	#607060
*APP*			0.0240	54.42	561.1	high	#104300, #605714
*JUN*			0.0200	45.35	97.62	high	
*CFTR*			0.0189	42.86	0.9818	low	
*HTT*			0.0179	40.59	37.64	medium	#143100, #617435
*DISC1*	2	0	0.0169	38.32	2.495	low	#604906
*MYC*			0.0161	36.51	3.305	low	
*CUL3*	1	0	0.0150	34.01	22.88	medium	#619239
*EGFR*			0.0138	31.29	7.925	low	
*BMI1*			0.0105	23.81	50.56	high	
*GRB2*			0.0102	23.13	98.53	high	
*YWHAG*	3	1	0.0097	22.00	554.5	high	#617665
*MAPT*	3	0	0.0096	21.77	223	high	#600274, #172700, #601104, #260540, #168600
*HSP90AB1*			0.0090	20.41	928.2	high	
*MEOX2*			0.0087	19.73	0.6813	low	
*SNW1*			0.0083	18.82	64.66	high	
*HSCB*			0.0080	18.14	14.33	medium	
*SPRED1*			0.0075	17.01	19.18	medium	#611431
*HRAS*	1	0	0.0072	16.33	77.62	high	#218040
*HSPB1*			0.0066	14.97	164.3	high	#606595, #608634
*CANX*			0.0066	14.97	164	high	
*ATXN1*			0.0066	14.97	12.41	medium	#164400
*FGFR3*			0.0065	14.74	148.4	high	#616482
*YWHAZ*	3	0	0.0063	14.29	236.1	high	
*NEK4*			0.0059	13.38	15.81	medium	
*RYBP*			0.0059	13.38	18.68	medium	
*ERBB2*			0.0057	12.93	9.006	low	
*CDC5L*			0.0057	12.93	21.79	medium	
*HSPD1*			0.0057	12.93	141.3	high	

Note: input genes: SFARI score = 1 or 2, and syndromic = 0. ^a^ Calculated by dividing each betweenness centrality value by the maximum betweenness centrality value in network A. ^b^ Retrieved from Genotype-Tissue Expression (GTEx) Portal (transcript per million, TPM). ^c^ high: >50 TPM; medium: 10–50 TPM; low: <10 TPM. ^d^ Restricted to neurological phenotypes (last update: July 2024).

**Table 2 ijms-26-02078-t002:** Top 30 genes in CNVs mapped on network A.

Gene	SFARI Score	Syndromic	Betweenness Centrality	Relative Betweenness Centrality ^a^ (%)	Expression in Brain ^b^ (TPM)	Brain Expression ^c^	OMIM Phenotype ^d^
*MKI67*			0.00299	6.78	0.04925	low	
*RAC1*		1	0.00248	5.62	351.2	high	#617751
*PRKN*	2	0	0.00209	4.74	7.774	low	#600116
*SMARCB1*			0.00154	3.49	98.65	high	
*LMO2*			0.00118	2.68	27.56	normal	
*PRKAB2*			0.00114	2.59	38.85	normal	
*CEP128*			0.00108	2.45	1.317	low	
*MAPK3*	2	0	0.00104	2.36	123.1	high	
*F2RL1*			0.00103	2.34	1.243	low	
*KCNMA1*	2	0	0.00096	2.18	14.71	normal	#618596, #617643
*FGFR2*			0.00086	1.95	130.1	high	
*LGALS8*			0.00077	1.75	30.62	normal	
*HAX1*			0.00074	1.68	87.22	high	
*PTPN1*			0.00072	1.63	21.84	normal	
*NDN*			0.00071	1.61	95.52	high	
*NOTCH2NLA*			0.00070	1.59	0.4823	low	
*UBE3A*	1	1	0.00060	1.36	16.90	normal	#105830
*CRKL*			0.00057	1.29	71.85	high	
*MKRN3*			0.00054	1.22	3.434	low	
*PSMB1*			0.00054	1.22	77.33	high	#620038
*OIP5*			0.00053	1.20	0.7748	low	
*COMT*			0.00052	1.18	56.20	high	#181500
*GLRX3*			0.00044	1	17.09	normal	
*AKAP9*	2	0	0.00044	1	22.25	normal	
*HERC2*		1	0.00042	0.95	31.09	normal	#615516
*TPM3*			0.00039	0.88	58.94	high	
*CHRNA7*	2	0	0.00038	0.86	0.3922	low	
*ZMYND11*	2	0	0.00037	0.84	97.83	high	#616083
*MFHAS1*			0.00036	0.82	12.92	normal	
*GABRB3*	1	0	0.00036	0.82	21.47	normal	#617113

Note: input genes: SFARI score = 1 or 2, and syndromic = 0. ^a^ Calculated by dividing each betweenness centrality value by the maximum betweenness centrality value in network A. ^b^ Retrieved from Genotype-Tissue Expression (GTEx) Portal (transcript per million, TPM). ^c^ high: >50 TPM; medium: 10–50 TPM; low: <10 TPM. ^d^ Restricted to neurological phenotypes (last update: July 2024).

**Table 3 ijms-26-02078-t003:** Results of ORA on the lists of genes within CNVs mapped on network A.

Input List	Pathway	Genes	FDR	ER
All	22q11.2 deletion syndrome ^a^	*FGFR2*, *CRKL*, *COMT*, ***LZTR1***, ***CLTCL1***, *PI4KA*, *SNAP29*, *THAP7*, *DGCR8*, *TRMT2A*, *GSC2*, *KLHL22*, *SLC25A1*, *C22orf39*, *DGCR6*, *ESS2*, *UFD1*, *MED15*, *MRPL40*, *HIRA*, *GP1BB*, *CLDN5*, *DGCR6L*, *ARVCF*, *TSSK2*, *RANBP1*, ***GNB1L***, *CDC45*, *HIRIP3*, *SCARF2*, *RTN4R*, *HES1*, *RTL10*, *SERPIND1*, *AIFM3*, *SLC7A4*	<2.2 × 10^−16^	13.0
	Cannabinoid receptor signaling ^a^	***MAPK3***, *MAPK8*, ***ADORA2A***, *PRKACG*, *DAGLB*, *MAPK10*, *AHR*	0.00499	7.5
	Prader–Willi and Angelman syndromes ^a^	*NDN*, ***UBE3A***, *MKRN3*, ***HERC2***, ***GABRB3***, ***CYFIP1***, *SNRPN*, ***TUBGCP5***, ***NIPA2***, *OCA2*, ***NIPA1***, ***ATP10A***, *GABRA5*, ***GABRG3***	0.0000014	7.1
Top 80%	22q11.2 deletion syndrome ^a,^*****	*FGFR2*, *CRKL*, *COMT*, ***LZTR1***, ***CLTCL1***, *PI4KA*, *SNAP29*, *THAP7*, *DGCR8*, *TRMT2A*, *GSC2*, *KLHL22*, *SLC25A1*, *C22orf39*, *DGCR6*, *ESS2*, *UFD1*, *MED15*, *MRPL40*, *HIRA*, *GP1BB*, *CLDN5*, *DGCR6L*, *ARVCF*, *TSSK2*, *RANBP1*, ***GNB1L***, *CDC45*, *HIRIP3*, *SCARF2*, *RTN4R*, *HES1*	<2.2 × 10^−16^	13.3
	Cannabinoid receptor signaling ^a^	***MAPK3***, *MAPK8*, ***ADORA2A***, *PRKACG*, *DAGLB*, *MAPK10*	0.03237	7.4
	Prader–Willi and Angelman syndromes ^a,^*****	*NDN*, ***UBE3A***, *MKRN3*, ***HERC2***, ***GABRB3***, ***CYFIP1***, *SNRPN*, ***TUBGCP5***	0.04690	4.7
Top 50%	Ubiquitin-mediated proteolysis ^b^	***PRKN***, ***UBE3A***, ***HERC2***, ***UBR5***, ***BIRC6***, *ITCH*, *MID1*, *FANCL*, *SMURF2*, *PIAS3*	0.01269	4.8
	22q11.2 deletion syndrome ^a,^*****	*FGFR2*, *CRKL*, *COMT*, ***LZTR1***, ***CLTCL1***, *PI4KA*, *SNAP29*, *THAP7*, *DGCR8*, *TRMT2A*, *GSC2*, *KLHL22*, *SLC25A1*, *C22orf39*, *DGCR6*, *ESS2*, *UFD1*, *MED15*, *MRPL40*, *HIRA*, *GP1BB*	<2.2 × 10^−16^	12.4
	Prader–Willi and Angelman syndromes ^a^	*NDN*, ***UBE3A***, *MKRN3*, ***HERC2***, ***GABRB3***, ***CYFIP1***, *SNRPN*, ***TUBGCP5***	0.00601	6.6
Top 20%	FCERI-mediated MAPK activation	***RAC1***, ***MAPK3***, *MAPK8*, *GRAP2*	0.04033	19.7
	Neutrophil degranulation ^c^	***RAC1***, *PSMB1*, *ATP6AP2*, *MVP*, *CAT*, ***ILF2***, *S100A7*, *SNAP29*, *PECAM1*, *ALDOA*, ***CYFIP1***, *PSMD7*	0.04033	3.9
	22q11.2 deletion syndrome ^a,^*****	*FGFR2*, *CRKL*, *COMT*, ***LZTR1***, ***CLTCL1***, *PI4KA*, *SNAP29*, *THAP7*, *DGCR8*	0.00006	10.8
	Prader–Willi and Angelman syndromes ^a^	*NDN*, ***UBE3A***, *MKRN3*, ***HERC2***, ***GABRB3***, ***CYFIP1***, *SNRPN*	0.00045	11.8

Note: SFARI genes are indicated in bold. FDR = false discovery rate; ER = enrichment ratio. ^a^ = Wikipathway, ^b^ = KEGG, ^c^ = Reactome, * = significantly enriched in the random ORA.

**Table 4 ijms-26-02078-t004:** Results of ORA performed on network B.

Pathway	Genes	FDR	ER
RNA polymerase ^b^	*POLR2B*, *POLR3C*, *POLR3E*, *POLR3F*	0.034	12.0
Fc epsilon RI signaling pathway ^b^	*ALOX5*, *MAPK10*, ***MAPK3***, *MAPK8*, ***RAC1***	0.042	6.9
Salmonella infection ^b^	*MAPK10*, ***MAPK3***, *MAPK8*, ***RAC1***, *RILP*, *TJP1*	0.034	6.5
Progesterone-mediated oocyte maturation ^b^	*KIF22*, *MAD1L1*, *MAPK10*, ***MAPK3***, *MAPK8*, *PRKACG*	0.042	5.6
Tight junction ^b^	***DLG2***, *MAPK10*, *MAPK8*, *PPP2R2D*, *PRKAB2*, *PRKACG*, ***RAC1***, *TJP1*, *TJP2*	0.025	4.9
Ubiquitin-mediated proteolysis ^b^	***BIRC6***, *FANCL*, ***HERC2***, *ITCH*, ***PRKN***, ***UBE3A***, ***UBR5***	0.042	4.8
FCERI-mediated MAPK activation ^c^	*GRAP2*, *MAPK10*, ***MAPK3***, *MAPK8*, ***RAC1***	0.046	13.9
22q11.2 deletion syndrome ^a^	*AIFM3*, *C22orf39*, *DGCR6*, *DGCR6L*, *DGCR8*, *ESS2*, *FGFR2*, *GSC2*, *HIRA*, ***LZTR1***, *MRPL40*, *PI4KA*, *SLC25A1*, *THAP7*, *TRMT2A*	1.63 × 10^−10^	12.8
Prader–Willi and Angelman syndromes ^a^	***CYFIP1***, ***HERC2***, *MKRN3*, *NDN*, *SNRPN*, ***TUBGCP5***, ***UBE3A***	0.0043	8.4

Note: SFARI genes are indicated in bold. FDR = false discovery rate; ER = enrichment ratio. ^a^ = Wikipathway, ^b^ = KEGG, ^c^ = Reactome.

## Data Availability

The raw data supporting the conclusions of this article will be made available by the authors on request.

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
