# Peer review of "A Systems Biology Approach for Prioritizing ASD Genes in Large or Noisy Datasets"

_ijms, 2025, doi:10.3390/ijms26052078_

Round 1

Reviewer 1 Report

Comments and Suggestions for Authors

This paper presents a systems biology approach to identify and prioritize genes associated with Autism Spectrum Disorder (ASD), aiming to uncover new candidates and elucidate the complex genetic landscape of the disorder. The authors constructed a Protein-Protein Interaction (PPI) network from known ASD-associated genes and utilized betweenness centrality to prioritize genes. This approach was aplied to a list of genes within copy number variants of unknown significance in 135 ASD patients.

This is a commendable effort to elucidate the complex genetic landscape of Autism Spectrum Disorder (ASD) through a systems biology approach. The major shortcoming is centered on the use of network centrality measures. I would suggest a more explicit discussion on the use of centrality measures in this context. While the authors acknowledge that these metrics identify hubs in the PPI network, which may not necessarily correspond to the proteins most relevant to the disorder, a more detailed discussion to clarify the limitations and assumptions of this approach could strengthen the manuscript and provide a clearer understanding of the results and their implications.

Validation on other experimental datasets would further strengthen the paper, but I understand that is not always possible.

Overall, I believe the paper makes a valuable contribution to the field and recommend it for publication.

Author Response

Reviewer #1

This paper presents a systems biology approach to identify and prioritize genes associated with Autism Spectrum Disorder (ASD), aiming to uncover new candidates and elucidate the complex genetic landscape of the disorder. The authors constructed a Protein-Protein Interaction (PPI) network from known ASD-associated genes and utilized betweenness centrality to prioritize genes. This approach was aplied to a list of genes within copy number variants of unknown significance in 135 ASD patients.

This is a commendable effort to elucidate the complex genetic landscape of Autism Spectrum Disorder (ASD) through a systems biology approach. The major shortcoming is centered on the use of network centrality measures. I would suggest a more explicit discussion on the use of centrality measures in this context. While the authors acknowledge that these metrics identify hubs in the PPI network, which may not necessarily correspond to the proteins most relevant to the disorder, a more detailed discussion to clarify the limitations and assumptions of this approach could strengthen the manuscript and provide a clearer understanding of the results and their implications.

Validation on other experimental datasets would further strengthen the paper, but I understand that is not always possible.

Overall, I believe the paper makes a valuable contribution to the field and recommend it for publication.

Thank you very much for your thoughtful feedback and constructive suggestions. We truly appreciate your recognition of the effort to uncover the genetic landscape of ASD through our systems biology approach. As per your suggestion, we have expanded the discussion on the use of betweenness centrality, providing more clarity on its limitations and assumptions in the context of ASD. We hope this addition helps strengthen the manuscript and provides a clearer understanding of the methodology and its implications.

Regarding validation on other experimental datasets, we fully agree that this would add valuable insight. While we are unable to include such data at this stage, we plan to pursue this direction in future studies if additional datasets become available. Once again, thank you for your thoughtful review and support of our work.

Reviewer 2 Report

Comments and Suggestions for Authors

Remori and coworkers developed a System biology approach to uncover new candidate genes for ASD, a system based on a protein-protein interaction (PPI) network generated from genes associated with ASD present in a public database.

The proposed System, although interesting, presents, in my opinion, some critical points:

  • Lack of specificity. The PPI network is too large and includes a very high fraction of human genes, which probably raises doubts about its real ability to highlight new ASD-relevant genes.
  • A more selective filter in the identification of early interactors is missing. As an example, the level of functional interactions in the central nervous system.
  • The use of betweenness centrality is interesting, but it is probably not sufficient if not integrated with gene expression analysis in brain tissues or with the presence of de novo mutations in ASD patients.

Some details:

The PPI network obtained presents more than 12,000 nodes, which represent about 63% of human coding genes. A number so high that it would suggest that the network is very large and not very specific. It appears that the system has recovered more than half of the genes as prime interactors of SFARI genes, possibly indicating that many of these interactors are not relevant to ASD, but only high connected in the protein interaction network, making the results obtained not very useful for identifying new ASD-specific genes.

Considering that SFARI is already a database that collects candidate genes for ASD, it is likely that many interactors are simply ubiquitous and highly connected genes, rather than actually involved in ASD. Approximately 80% of the SFARI genes present in network A show physical interactions, but this does not necessarily imply a causal relationship.  I think it is useful, if not necessary, to extend the analysis considering only interations in brain contexts or during neuronal development.

Betweenness centrality is very useful but tends to highlight highly connected hubs, not necessarily relevant to ASD. I think it is more informative to combine betweenness centrality with other functional data, such as spatiotemporal expression in ASD-relevant tissues, as previusly said.

The enrichment analysis (ORA) show pathways that seem to emerge even in the random lists. 22q11.2 Deletion Syndrome, Prader-Willi and Angelman seem significantly enriched even in the randomly selected gene sets. This seems to indicate that these pathways are intrinsically overrepresented in the initial dataset and not necessarily linked to ASD specifically. Pathways such as Salmonella infection, Progesterone-mediated oocyte maturation, RNA polymerase and Tight junction could also have an indirect link with neurodevelopment, but overall they raise doubts about the specificity of the results.

The arrayCGH highlights only CNVs, deletions and duplications of DNA tracts, does not provide information on gene expression and says nothing about variations present in nervous tissue. It would be more informative to have gene expression data in ASD brain tissue or at least in some cellular models.

I suggest, to make the system usable:

  • further filter genes based on expression in brain tissues and neuronal development
  • use functional data (de novo mutations, RNA-seq of brain tissue) to validate the prioritized genes
  • check whether the identified genes show co-expression patterns with already known ASD genes

Author Response

Reviewer #2

Remori and coworkers developed a System biology approach to uncover new candidate genes for ASD, a system based on a protein-protein interaction (PPI) network generated from genes associated with ASD present in a public database.

The proposed System, although interesting, presents, in my opinion, some critical points:

Lack of specificity. The PPI network is too large and includes a very high fraction of human genes, which probably raises doubts about its real ability to highlight new ASD-relevant genes.

A more selective filter in the identification of early interactors is missing. As an example, the level of functional interactions in the central nervous system.

The use of betweenness centrality is interesting, but it is probably not sufficient if not integrated with gene expression analysis in brain tissues or with the presence of de novo mutations in ASD patients.

Some details:

The PPI network obtained presents more than 12,000 nodes, which represent about 63% of human coding genes. A number so high that it would suggest that the network is very large and not very specific. It appears that the system has recovered more than half of the genes as prime interactors of SFARI genes, possibly indicating that many of these interactors are not relevant to ASD, but only high connected in the protein interaction network, making the results obtained not very useful for identifying new ASD-specific genes.

Considering that SFARI is already a database that collects candidate genes for ASD, it is likely that many interactors are simply ubiquitous and highly connected genes, rather than actually involved in ASD. Approximately 80% of the SFARI genes present in network A show physical interactions, but this does not necessarily imply a causal relationship.  I think it is useful, if not necessary, to extend the analysis considering only interations in brain contexts or during neuronal development.

Betweenness centrality is very useful but tends to highlight highly connected hubs, not necessarily relevant to ASD. I think it is more informative to combine betweenness centrality with other functional data, such as spatiotemporal expression in ASD-relevant tissues, as previusly said.

The enrichment analysis (ORA) show pathways that seem to emerge even in the random lists. 22q11.2 Deletion Syndrome, Prader-Willi and Angelman seem significantly enriched even in the randomly selected gene sets. This seems to indicate that these pathways are intrinsically overrepresented in the initial dataset and not necessarily linked to ASD specifically. Pathways such as Salmonella infection, Progesterone-mediated oocyte maturation, RNA polymerase and Tight junction could also have an indirect link with neurodevelopment, but overall they raise doubts about the specificity of the results.

The arrayCGH highlights only CNVs, deletions and duplications of DNA tracts, does not provide information on gene expression and says nothing about variations present in nervous tissue. It would be more informative to have gene expression data in ASD brain tissue or at least in some cellular models.

I suggest, to make the system usable:

further filter genes based on expression in brain tissues and neuronal development

use functional data (de novo mutations, RNA-seq of brain tissue) to validate the prioritized genes

check whether the identified genes show co-expression patterns with already known ASD genes

  1. Specificity of the PPI Network and lack of a selective filter for early interactors

Thank you for your valuable feedback. To further demonstrate the specificity of our network, we conducted an additional analysis using a Monte-Carlo approach. In this analysis, we randomly sampled the same number of genes from the Human Genome Nomenclature Committee (HGNC) database using 1,000 different random seeds. Our results confirmed that the enrichment of SFARI genes in Network A is statistically significant (p < 2E-16) for all three categories of SFARI genes, further supporting the notion that the observed enrichment is not due to random chance.

Moreover, we took an additional step to refine the PPI network by incorporating gene expression data from the Human Protein Atlas. Specifically, we filtered interactions based on gene expression levels in 966 samples from the Human Brain Tissue Bank (HBTB), which were analyzed using RNA-seq on the MGI platform. This enabled us to focus on genes that are expressed in the brain. The filtering process resulted in a reduction of the network size to 11,879 nodes, which corresponds to 94.3% of the original network size. This additional step reinforces the sensitivity and specificity of our approach, as it successfully retained genes that are actively expressed in brain tissues and likely more relevant to ASD.

However, for our study’s objectives, we chose to retain the original, unfiltered network rather than the filtered version. The rationale behind this is that there may be genes expressed in the brain that have not yet been identified and could still play a crucial role in ASD. Our approach is focused on prioritizing the study of variants in highly connected genes, particularly those interacting with SFARI genes, rather than isolating single causative genes. By keeping the broader network, we aim to explore potential novel interactions that could lead to new insights into ASD-related pathways.

Moreover, this approach is particularly valuable for prioritizing genes in datasets where gene expression data is unavailable, such as CNVs identified through array-CGH analysis. This tool was designed with the goal of offering a system that can process noisy datasets like CNVs and prioritize relevant variants based on the connectivity of the genes within the PPI network, rather than relying solely on expression data. By keeping the broader network intact, we provide an essential resource for the discovery of novel genes, even in cases where expression data or other functional data may not be available.

We believe these results strengthen the relevance of the identified interactors in the context of ASD and address the concern that many of the interactors might be general, highly connected proteins not specific to ASD. Moreover, we believe that maintaining the broader, unfiltered network is essential for uncovering novel insights into ASD-related pathways. By not limiting the analysis to a narrow set of genes, we are enabling the identification of previously unrecognized interactions that might be crucial for understanding the complex biology of ASD. This approach is particularly valuable given the incomplete understanding of ASD’s genetic architecture. Highly connected genes, even if not directly linked to ASD in the past, may play key roles in larger, previously unexplored biological networks. Therefore, our broader network allows us to generate hypotheses that could lead to discovering new ASD-specific genes or mechanisms, which may otherwise remain hidden if we were to focus only on already well-characterized candidates.

To assess all your valuable points, we implemented the text adding the results of Monte-Carlo approach to confirm the specificity of Network A and the information on the percentage of nodes annotated in the Human Protein Atlas as brain expressed to validate the sensitivity of the network A in retrieving brain expressed genes.

  1. Use of betweenness centrality and integration with functional data (co-expression patterns)

Thank you for your comment. To investigate the relevance of the identified genes in the context of ASD, we analyzed gene expression data from 13 brain regions available in the Human Protein Atlas. These regions include the amygdala, basal ganglia, cerebellum, cerebral cortex, choroid plexus, hippocampal formation, hypothalamus, medulla oblongata, midbrain, pons, spinal cord, thalamus, and white matter. We then calculated the correlation between the expression of the top 30 genes identified in Table 1 and the 224 SFARI score 1 genes (those with high confidence in their association with autism) present in Network A. The results showed high correlation values between the SFARI genes and the top 30 genes, suggesting that these genes are co-expressed in key brain regions. However, we also observed that CUL3, a SFARI score 1 gene, showed low correlation with other SFARI genes. This finding highlights an important aspect of gene interaction in the context of ASD. We added this analysis in the text. While co-expression is a valuable measure for understanding functional relevance, it is not the sole factor for prioritizing genes involved in ASD. Our study focuses on the identification of ASD-related genes through betweenness centrality, a key network measure that identifies genes with critical positions in the protein-protein interaction (PPI) network. These genes may not necessarily be co-expressed with other ASD-related genes, but their centrality in the network suggests that they may play a pivotal role in the regulation of ASD-related pathways by serving as important bridges between different functional modules. Betweenness centrality allowed us to prioritize genes that are important due to their topological role in the network rather than their expression patterns. For researchers who do not have access to gene expression data, this approach offers a valuable tool to uncover potential ASD risk genes based purely on their network connectivity. By identifying genes that are crucial for the flow of information within the PPI network, we can predict potential ASD-related genes even without direct expression data. Thus, the use of betweenness centrality in our study highlights its utility for prioritizing genes based on their topology in the network, which is particularly relevant for cases where only genetic variant information is available.

This also underscores the utility of betweenness centrality for prioritizing genes in datasets such as CNVs, where expression data is not available. By combining network topology with genetic data, we are able to provide a robust framework for identifying potentially relevant genes from noisy datasets like CNVs, which may otherwise remain challenging to interpret.

  1. Specificity of pathway enrichment in ORA results

Thank you for your insightful comment regarding the enrichment analysis and the pathways that emerged, such as 22q11.2 deletion syndrome, Prader-Willi syndrome, and Angelman syndrome, as well as other pathways like Salmonella infection and cannabinoid receptor signaling. We appreciate your concern regarding the specificity of these findings. We agree that the enrichment of developmental syndromes like 22q11.2 deletion syndrome, Prader-Willi syndrome, and Angelman syndrome in our analysis raises important questions about the potential overlap between ASD and other neurodevelopmental disorders. However, these findings are not entirely unexpected. For example, the 22q11.2 deletion syndrome is well-known to be associated with a range of neurodevelopmental phenotypes, including ASD, and several studies have linked ASD to mutations in genes associated with these syndromes (e.g., UBE3A in Angelman syndrome). Thus, the identification of these pathways in our analysis is consistent with existing knowledge about the genetic overlap between ASD and these developmental disorders.

In this study, we used the CNV gene list as a proof of concept to demonstrate how our network, built from SFARI-associated genes, can be employed to prioritize genes in a noisy dataset, such as CNVs. The observed enrichment in pathways linked to ASD supports our hypothesis that mapping genes from such a noisy dataset onto a PPI network can help prioritize potentially relevant variants. While the identification of pathways such as 22q11.2 deletion syndrome and Prader-Willi syndrome may seem to reflect general neurodevelopmental syndromes, the fact that these pathways emerged from the top-ranking genes, those with high betweenness centrality, indicates that the most central genes in the network are associated with highly relevant ASD-related pathways. This suggests that our approach is effective in identifying genes that play a significant role in ASD.

Furthermore, to control for dataset biases, we generated random gene lists and performed the ORA on those as well. The enrichment of these pathways in the random lists, particularly 22q11.2 deletion syndrome, further supports the notion that these pathways are not artifacts of our methodology but rather an inherent feature of the dataset. The fact that these pathways emerged consistently across both ranked by centrality and random gene sets reinforces the idea that mapping CNV genes to our PPI network helps prioritize variants that might otherwise be overlooked.

Regarding pathways like Salmonella infection and cannabinoid receptor signaling, we agree that they may not appear directly relevant to ASD at first glance. However, it is important to note that many pathways involved in neurodevelopment and immune regulation, such as those modulated by the cannabinoid receptor system, have been shown to affect ASD-related phenotypes. For example, the endocannabinoid system has been implicated in synaptic plasticity and neurodevelopment, and studies suggest its potential role in modulating social behaviors and anxiety-related symptoms, which are common in ASD. Similarly, the Salmonella infection pathway, while seemingly unrelated to ASD, could reflect broader immune and inflammatory responses that are known to influence neurodevelopmental disorders.

In summary, while some of the enriched pathways may appear to be general or unrelated to ASD, we believe they warrant further investigation due to their potential indirect involvement in neurodevelopmental processes. Moreover, we acknowledge that a more refined focus on ASD-specific pathways could be considered in future analyses, particularly if gene expression data from brain tissues or cellular models were available. However, these future analyses are out of scope of the present work, which focuses on providing a valuable tool to prioritize sequence and copy number variants when expression data are unavailable.

Thank you again for your valuable suggestions, and we hope this response clarifies our approach and the relevance of the identified pathways in the context of ASD. To take advantage while replying to your helpful concerns, we modified the discussion section accordingly.

Round 2

Reviewer 2 Report

Comments and Suggestions for Authors

The authors' response was very thorough and the manuscript was adequately improved. I believe this version can be accepted for publication in IJMS.